# Quaternary Ammonium Salts Interact with Enolates and Sulfonates via Formation of Multiple +N-C-H Hydrogen Bonding Interactions

Grazia Bencivenni [1], Nathalie Saraiva Rosa [1], Paolo Grieco [2], Malachi W. Gillick-Healy [3], Brian G. Kelly [3], Brendan Twamley [4] and Mauro F. A. Adamo [1,*]

[1] Department of Chemistry, RCSI, University of Medicine and Health Science, 123 St Stephen's Green, D02 Dublin, Ireland; grazia.bencivenni@gmail.com (G.B.); nathaliesaraivarosa@rcsi.ie (N.S.R.)

[2] Faculty of Pharmacy, University of Naples Federico II, Corso Umberto I, 40, 80138 Napoli, NA, Italy; paolo.grieco@unina.it

[3] KelAda Pharmachem Ltd., A1.01, Science Centre South, Belfield, D04 Dublin, Ireland; malachi.gh@keladapharmachem.com (M.W.G.-H.); brian.kelly@keladapharmachem.com (B.G.K.)

[4] School of Chemistry, Trinity College Dublin, University of Dublin, D02 Dublin, Ireland; twamleyb@tcd.ie

[*] Correspondence: madamo@rcsi.ie

**Abstract:** We report herein sharp physical evidence, i.e., single-crystal X-ray diffraction and $^1$H-NMR spectral data, confirming that quaternary ammonium species interact with anions via a set of directional ion–dipole cooperative +N-C-H unusual H-bonding interactions and not via pure non-directional ionic electrostatic interactions. This finding, which has been often invoked by calculations, is herein substantiated by the preparation of two model compounds and an analysis of their X-ray crystal structures in the solid state and $^1$H-NMR spectra in solution. These observations are particularly pertinent for the rational design of novel catalyses and catalysts and providing guidance to an understanding of these species in solution and during asymmetric enantioselective catalysis.

**Keywords:** quaternary ammonium salts; phase-transfer catalysis; unusual H-bond



## 1. Introduction

Quaternary ammonium salts are a pivotal class of organocatalysts that have been discovered and used in chemical synthesis for the formation of C-C, C-N, and C-O bonds for half a century, thus cementing their roles as essential tools in the pharmaceutical, agrochemical, and fine chemical industries [1–3]. In particular, quaternary ammonium salts derived from *Cinchona* alkaloids [4,5] took the stage versus other organocatalysts as they are simple to prepare, easy to recover, and usually provide a higher turnover when compared to other catalyses. While many studies focused on the applications of these species in asymmetric synthesis, only a few reports have dealt with the mechanistic features underlying the mode(s) of interaction between the quaternary ammonium ion and the reagents [6–8]. In order to confirm the nature of molecular recognition exerted by chiral quaternary ammonium ions and neutral organic reagents, we carried out a set of $^1$H-NMR titration studies that demonstrated that +N-C-H protons do participate in hydrogen bonding with oxygen atoms from functional groups such as -NO$_2$, R$_1$R$_2$C=O, and R$_1$R$_2$NC=O, when those functional groups are present in neutral molecules [9]. Hence, we set out to demonstrate that a set of non-classical, yet directional, H-bonds govern the enantioselective outcome of reactions when both *Cinchona*- and Maruoka-type catalysts are employed. With this result in mind, we then posed the question of determining which type of interaction occurs between quaternary ammonium salts and fully formed anions. Quaternary ammonium salts are often involved with the chemistry of enolates and investigating whether their interactions occur (1) via ionic bonding or (2) via +N-C-H H-bonding. Given that ionic bonds are non-directional and hydrogen bonds are directional, it is important to

understand which interactions are operative when discussing the preferential formation of one enantiomer versus another. In this regard, deciphering the exact mode of the molecular recognition of chiral ammonium species and their ionic substrates would provide an indispensable tool for the design of new catalysts. The existence of an electrostatic interaction between quaternary ammonium salts and fully formed anions has been shown to operate in the solid state via the X-ray crystal structures of tetra-*n*-butylammonium alkoxides [10,11] and of pentafluorobenzyl-substituted ammonium and pyridinium salts [12]. Hence, at least at the solid state, this electrostatic interaction was revealed to be an ion–dipole interaction formed by $^+$N-C-H and RO$^-$ moieties. Corey [13] and Jørgensen [14] reported the X-ray crystal structures of two *Cinchona*-based quaternary ammonium salts with *p*-nitrophenolate, allegedly pointing out that an ionic interaction was formed by the phenoxide and the ammonium in the crystal cage (vide infra). However, Reetz showed by molecular orbital calculations that the charge in the case of the tetrabutylammonium unit is not localized on the nitrogen, but rather it is delocalized over the adjacent four methylene groups, implicating that α-methylene groups in quaternary ammonium salts are acidic [15] and are, hence, able to act as H-bond donors. This finding was further confirmed by Shirakawa and co-workers who prepared a quaternary ammonium salt containing an electron-withdrawing group and demonstrated the acidity of this species in the catalysis of the Mannich reaction [16]. In summary, an analysis of the current literature pointed out that quaternary ammonium salts may interact (1) with neutral organic molecules via the $^+$N-C-H hydrogen bond [9] and (2) with charged anions via ion–dipole interactions, rather than an ionic bond. However, there is a gap in the literature that reports real physical data showing that, when in solution, quaternary ammonium salts interact with anions via a directional H-bond. In order to bridge this gap, we designed compounds **1** and **2** (Figure 1), in which an ammonium ion is linked intra-molecularly with a sulfonate anion or an enolate, respectively. Compounds **1** and **2** were synthesized and their NMR spectra and X-Ray data were used to confirm the engagement of $^+$N-C-H moieties in anion binding.

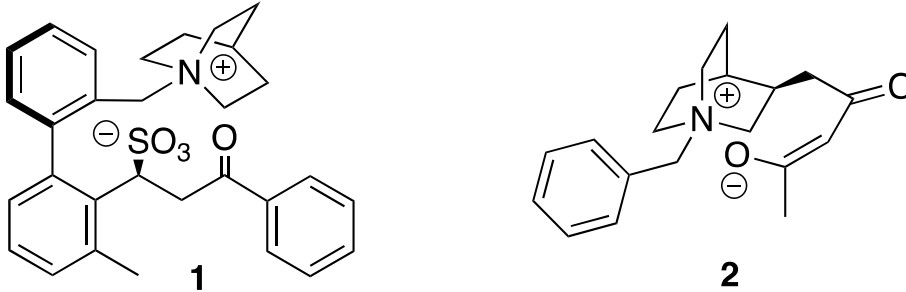

**Figure 1.** Probes (**1**) and (**2**) used in this study.

## 2. Results and Discussion

In light of the results we recently reported [9], which demonstrated using physical data that -NO$_2$ moieties are exceptionally good ligands for quaternary ammonium salts, we had cause to re-evaluate data reported by Corey and co-workers [13]. In the seminal paper, they reported the X-ray data of *O*(9)-allyl-*N*-(9-anthracenylmethyl)-cinchonidinium *p*-nitrophenoxide and provided a rationalization for the interaction of *Cinchona*-based quaternary ammonium species and *p*-nitrophenoxide as an ionic ammonium-to-alkoxide interaction. Given that observations made from our data seemed to challenge the interpretations reported by Corey and co-workers, we decided to re-examine the X-ray data they deposited in the CCDC database (Figure 2), for which we would like to provide an alternative interpretation. Firstly, the fundamental unit at a solid state is formed by a molecule of *O*(9)-allyl-*N*-(9-anthracenylmethyl)-cinchonidinium and its counter ion *p*-nitrophenoxide; however, the H-bond (2.24 Å) occurring between the -NO$_2$ oxygen and the benzylic $^+$N-C-H proton was the shortest interaction (Figure 2A). It should also be noted that oxygen atoms from both the nitro (-NO$_2$) and aryloxy (ArO$^-$) moieties of *p*-nitrophenoxide can be

seen to interact with two molecules of *O*(9)-allyl-*N*-(9-anthracenylmethyl)-cinchonidinium (Figure 2B,C). If the latter were to be considered a purely electrostatic ionic bond, the distance between the centroids of the ions (3.45 Å) justifies an interpretation by which *p*-nitrophenoxide and *O*(9)-allyl-*N*-(9-anthracenylmethyl)-cinchonidinium interacted in the solid state primarily by their $^+$N-C-H and -NO$_2$ moieties, i.e., via a hydrogen bond. This is in accordance with what we demonstrated to take place in the solution, where direct physical evidence was shown for the H-bonding interaction of a -NO$_2$ oxygen and a benzylic $^+$N-C-H proton [9]. From the analysis of Corey's X-ray crystal structure (Figure 3C), the interaction of the ArO$^-$ moiety with the quaternary ammonium salt can be seen as two $^+$N-C-H (2.36 Å and 2.42 Å, respectively) ion–dipole interactions. Given our reinterpretation of the X-ray crystal structure of *O*(9)-allyl-*N*-(9-anthracenylmethyl)-cinchonidinium *p*-nitrophenoxide, it therefore becomes apparent that (1) the oxygen of the -NO$_2$ moiety engages in stronger binding with *O*(9)-allyl-*N*-(9-anthracenylmethyl)-cinchonidinium when compared to the oxygen of the ArO$^-$ and (2) the interaction between ArO$^-$ oxygen and quaternary ammonium salts should be interpreted as an ion–dipole interaction, rather than as an ionic bond. The nature of the interaction occurring between $^+$N-C-H donors and ions is relevant when considering that ion–dipole interactions are <u>directional</u>, for which reason they can be invoked to explain the origin of stereoselectivity.

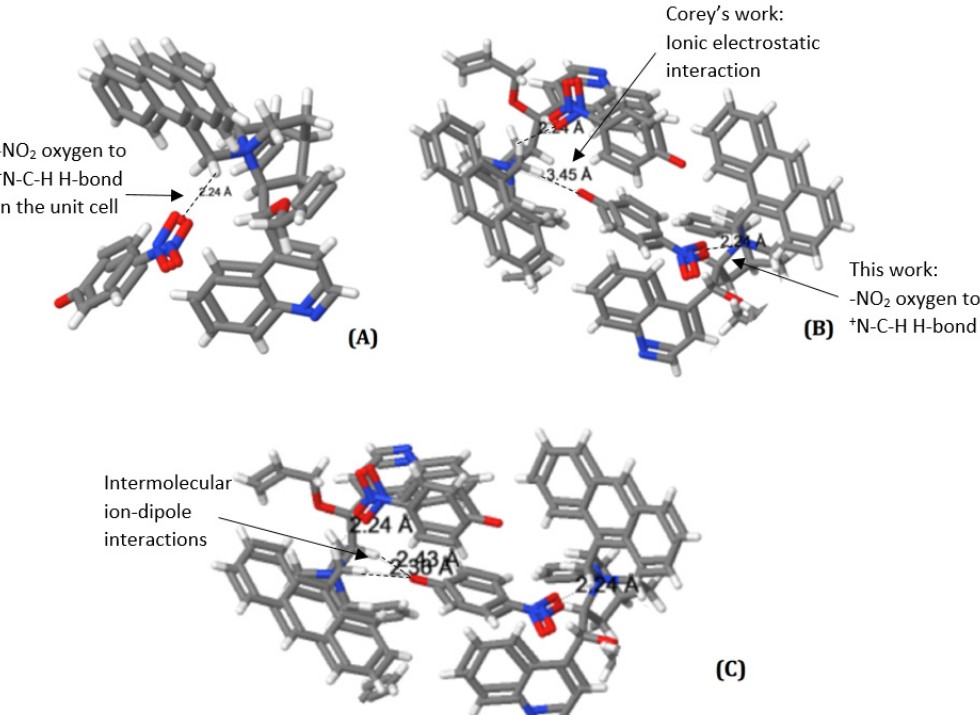

**Figure 2.** Re-evaluation of Corey and co-workers' CCDC data for *O*(9)-allyl-*N*-(9-anthracenylmethyl)-cinchonidinium *p*-nitrophenoxide [13]. (**A**) Hydrogen bonding interaction between an $^+$N-C-H proton to nitro oxygen of *p*-nitrophenoxide. (**B**) Comparison of (i) Corey's alleged ionic electrostatic interaction based on observation of 3.45 Å distance between phenoxide oxygen and ammonium nitrogen, versus (ii) Our observation of a 2.24 Å hydrogen bond between an $^+$N-C-H proton and nitro oxygen of *p*-nitrophenoxide. (**C**) Our observation of multiple hydrogen bonds (2.24, 2.36, 2.43 Å) between $^+$N-C-H protons to both the phenoxide oxygen and nitro oxygen moieties of *p*-nitrophenoxide.

In summary, on the basis of the interpretation of the X-ray data provided herein (Figure 2), and in light of the behavior of quaternary ammonium salts observed in solution, it became apparent that the nature of the inter-molecular interaction between *O*(9)-allyl-*N*-(9-anthracenylmethyl)-cinchonidinium and *p*-nitrophenoxide should be interpreted as occurring through an H-bond between a -NO$_2$ oxygen to $^+$N-C-H protons. As a consequence, it should no longer be considered acceptable to use the X-ray data reported by Corey and co-workers to demonstrate that the interaction of quaternary ammonium salts and phenoxides is ionic in nature. On the contrary, the π-facial model [17,18] (Figure 3), rather than the tetrahedron model, [13] appeared to us as most appropriate to explain the origin of stereoselectivity in reactions such as the alkylation of glycine imides using chiral quaternary ammonium salts. The π-facial model [17,18], supported by density functional theory calculations, provides an interpretation of the nature of an interaction between cinchonidinium salts and enolates, which, rather than involving an ionic bond, invokes the formation of: (i) an ion–dipole bond between the oxy-anion and the $^+$N-C-H proton; and (ii) a $^+$N-C-H proton-to-π interaction with the enolate.

Ess concluded that whether the nature of the interaction between cinchonidinium salts and enolates was: (i) a non-directional ionic interaction [19]; or (ii) a set of directional interactions borne of $^+$N-C-H protons interacting with ions and π-clouds [18], the relative orientation of prochiral enolates and the chiral catalyst was the same. While both the tetrahedron model and π-facial model explain accurately the stereochemical outcome of the alkylation of glycinamides, only the π-facial model could provide a general working model for the planning of other catalyses via organic anions.

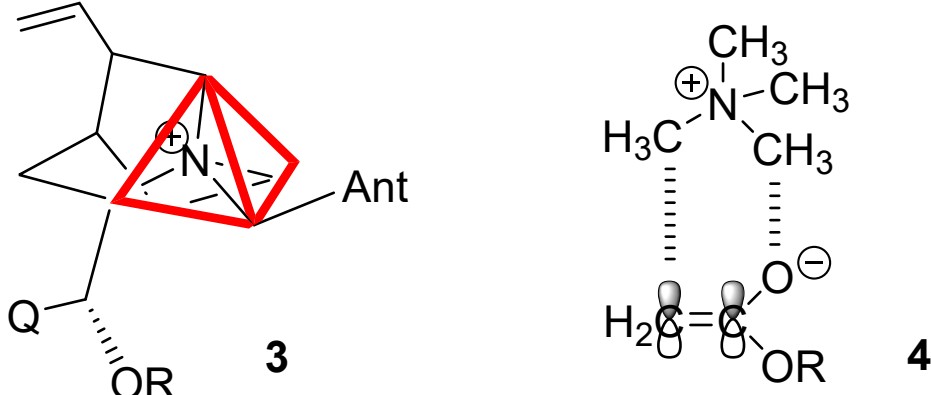

**Figure 3.** Tetrahedron model (**3**) and π-facial model (**4**) proposed for the explanation of the enantioselectivity of the alkylation of glycine enolates.

In order to confirm the nature of the interaction between quaternary ammonium salts and anions, we synthesized model compounds **1a** and **1b** (Scheme 1) bearing an ammonium group and an intra-molecular counter-ion.

Diastereoisomers **1a** and **1b** (Scheme 1) were obtained in five steps with an overall yield of 35%. Aldol condensation of aldehyde **5** and ketone **6** provided chalcone **7**, which, in turn, was reacted in a Suzuki coupling with boronic ester **8** using PEPPSI™-IPr ([1,3-Bis(2,6-Diisopropylphenyl)imidazol-2-ylidene](3-chloropyridyl)palladium(II) dichloride) as the catalyst [20]. The resulting **9** was then mesylated to obtain **10**, which was then reacted with quinuclidine **11** to provide quaternary ammonium species **12**. Compound **12** was then subjected to sulfonylation using the methodology previously developed by our group [21–23]. The sulfonylation of **12** gave two diastereoisomeric compounds, namely, **1a** and **1b** as a racemate, where one rotamer bears the benzylic -CH$_2$ and the sulfonate group on the same side with respect to the chiral axis of the molecule and a second one where they are on opposite sides (Scheme 1). Interestingly, the two diastereoisomers displayed distinct solubility profiles, where one was found to be more insoluble in methanol, while the other precipitated from dichloromethane. When separated, **1a** and **1b** were found to be

configurationally stable under heating and in the presence of acids and bases. Interestingly, the [1]H-NMR spectra of compounds **1a** and **1b** showed a remarkable enlargement of the signals for the quinuclidine $^+$NCH$_2$ protons, which supported the engagement of these CH moieties in H-bonding. This is noteworthy when considering that *Cinchona*-based quaternary ammonium salts supported on polystyrene sulfonate beads displayed the same enantioselectivity as when in solution [24]. This can only be explained by considering the $^+$N-CH$_2$ as the point of anchorage to the sulfonylated resin and not the ammonium, which is required for the catalysis. The X-ray diffractometry data for **1a** (Figure 4) confirmed the absolute configuration of the atropoisomeric biphenyl [25]. The X-ray crystal structure of **1a** showed an intra-molecular benzylic $^+$N-C-H proton to sulfonate H-bond of 2.2 Å in length and an inter-molecular interaction between the quinuclidinyl $^+$N-C-H proton to sulfonate of 2.4 Å length. Once again, the leading intra- and inter-molecular interactions observed in the crystal lattice from both (i) our X-ray diffractometry data of **1a** (Figure 4) and (ii) Corey's data of *O*(9)-allyl-*N*-(9-anthracenylmethyl)-cinchonidinium *p*-nitrophenoxide [13] (Figure 2) are directional and, therefore, should be interpreted as ion-to-dipole hydrogen bonding interactions. As discussed above, the nature of the interaction established between enolates and *Cinchona*-based quaternary ammonium salts during catalysis has been extensively studied and debated, with calculations supporting either an ionic bond [13,14,19] or an ion–dipole mode [17,18].

**Scheme 1.** Preparation of intra-molecular ammonium sulfonates (**1a**) and (**1b**).

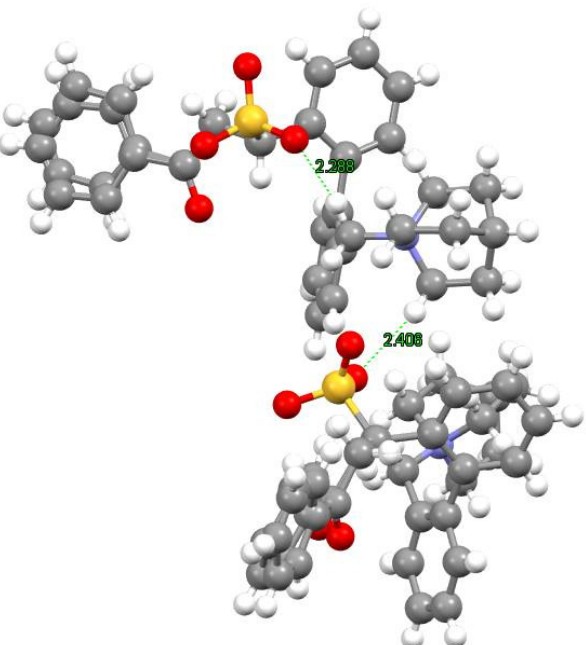

**Figure 4.** X-ray crystal structure of compound **1a**.

In order to probe the nature of the interaction between enolates and quaternary ammonium salts, we designed and prepared quinuclidine **2** whose structure was carefully engineered to permit clear observation of an intramolecular interaction between ammonium to enolate moeities. Our initial attempts to independently generate an enolate and to use this species for the $^1$H-NMR titration of *Cinchona*-based quaternary ammonium salts gave no significant results, for which reason compound **16** was designed specifically to permit the generation of compound **2** where the cation and the anion motifs are connected via an intra-molecular scaffold (Scheme 2). Our experimental plan was to generate ammonium enolate **2** in situ by treatment with a base, and intra-molecular H-bonding could then be confirmed by the shifting of spectral peaks as observed by solution-state $^1$H-NMR spectroscopy.

**Scheme 2.** Preparation of **2**.

The first step in the synthesis involved initial trans-esterification of **13** with quinuclidi-nol **14** to give ester **15** (Scheme 2). Ester **15** was then quaternarized with benzyl bromide to provide ammonium salt **16**, which was then dissolved in CD$_2$Cl$_2$ and treated with two equivalents of a proton sponge (1,8-bis(dimethylamino)naphthalene). The $^1$H-NMR spectra

were recorded over 24 h at intervals of 10 min and compared with the spectrum of pure **16** (Figure 5, see also Figure S14 in Supplementary Materials).

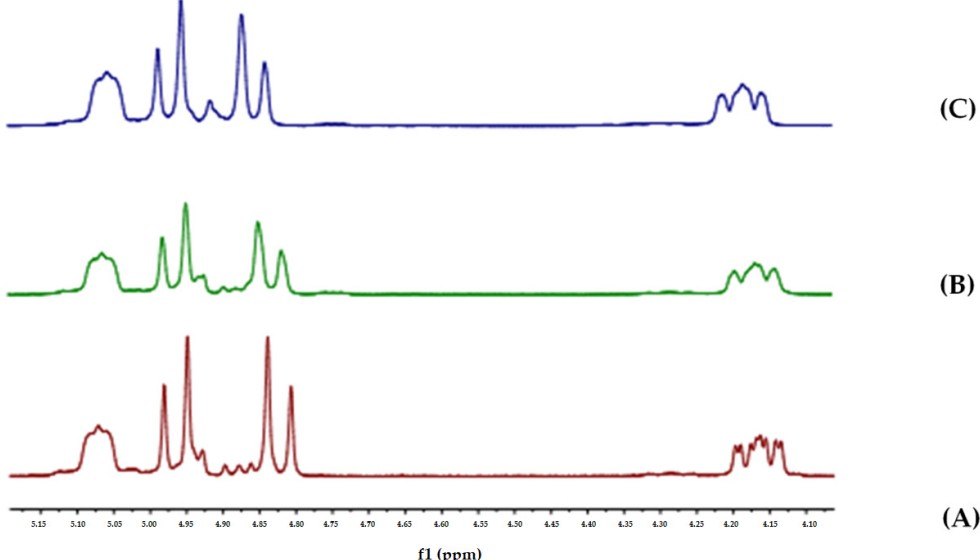

**Figure 5.** Selected $^1$H-NMR spectral region of compound **16** before and following the addition of base. (**A**) Compound **16**; (**B**) compound **16** and 2 equiv. of proton sponge after 10 min; (**C**) compound **16** and 2 equiv. of proton sponge after 24 h.

The relative integration of the $^1$H-NMR signals (the $CH_2$ between the two carbonyls in **16** integrated for 2 protons in the starting material and 1.83 protons after 24 h) indicated that under these conditions only a small amount of **16** was enolized to **2** (5%). Nevertheless, this experiment revealed that two $^+$N-C-H protons were significantly shifted upon the addition of the base, namely, one of the two benzylic $CH_2$ protons and one proton of the quinuclidinium ring. The affected protons, $H_1$, $H_2$, and $H_3$, were assigned via HSQC and HMBC 2D NMR experiments (Figure 6, see also Figures S16 and S17 in Supplementary Materials). In particular, a resonance in the HMBC spectrum was visible between the benzylic $H_1$ and three carbons at 59, 54, and 52 ppm, which were assigned to $C_1$, $C_2$, and $C_3$, respectively. A resonance in the HSQC spectrum observed between $C_1$ and two protons at 4.28 ppm and 3.86 ppm with the coupling constants of 12 Hz and 8 Hz, respectively, led us to assign $H_2$ as the equatorial proton linked to $C_1$. Proton $H_3$ was identified at 5.16 ppm via the analysis of the HMBC spectrum in which there was a resonance between the carbonyl ester carbon at 176 ppm and the proton at 5.16 ppm.

**Figure 6.** Selected $^1$H and $^{13}$C NMR signals for **2** assigned via HMBC and HSQC.

The data shown in Figures 5 and 6 undoubtedly point to an ion–dipolar interaction between the oxygen of the enolate moiety and the proton of the $^+$N-C-H moiety of quaternary

ammonium salt **2** in solution, which, to the best of our knowledge, has not been described in the literature to date.

## 3. Materials and Methods

### 3.1. General Experimental

NMR experiments were performed on a Bruker® Avance™ III 400 instrument, and samples were obtained in chloroform-d (CDCl$_3$) referenced to 7.26 ppm for [1]H and 77.16 for [13]C. Coupling constants ($J$) are in Hz. Multiplicities are reported as follows: s (singlet), d (doublet), dd (doublets of doublets), t (triplet), q (quartet), m (multiplet), c (complex), and br (broad). Mass spectra were recorded on a Waters® Micromass® LCT spectrometer using electrospray (ES) ionization techniques. All reagents and solvents were used as purchased from Sigma-Aldrich unless otherwise stated. Reactions were monitored for completion by thin-layer chromatography (TLC) (EM Science, silica gel 60 F254). Flash chromatography was performed using silica gel 60 (0.040–0.063 mm, 230–400 mesh) or alumina (activated, neutral, Brockmann activity I). The enantiomeric excess (ee) of the products was determined by chiral stationary phase HPLC (Daicel Chiralpak AD) using a UV detector operating at 217 nm and 227 nm. Infrared (IR) spectra were recorded as thin films between NaCl plates using a Bruker Tensor27 FT-IR instrument. Absorption maximum (V$_{max}$) was reported in wave numbers (cm$^{-1}$) and only selected peaks were reported.

### 3.2. Preparation of (E)-3-(2-Bromophenyl)-1-phenylprop-2-en-1-one (**7**)

Compound **7** was synthesized according to the literature [26]. Compound **7** was obtained as a yellow solid (1.43 g, 85%). [1]H NMR (400 MHz, CDCl$_3$) δ 8.13 (d, *J* = 16 Hz, 1H), 8.05–7.99 (m, 2H), 7.74 (dd, *J* = 8, 1 Hz, 1H), 7.66–7.56 (m, 2H), 7.51 (t, *J* = 7 Hz, 2H), 7.43 (d, *J* = 16 Hz, 1H), 7.36 (t, *J* = 7 Hz, 1H), 7.28–7.22 (m, 1H). [13]C NMR (101 MHz, CDCl3) δ 190., 143.14, 137.8, 135.0, 133.5, 132.9, 131.3, 128.6, 128.6, 127.8, 127.7, 125.8, 125.0.

### 3.3. Preparation of (E)-3-(2′-(Hydroxymethyl)-[1,1′-biphenyl]-2-yl)-1-phenylprop-2-en-1-one (**9**)

Potassium hydroxide (KOH) (134.4 mg, 1.2 equiv.) and 2-(hydroxymethyl)phenylboronic acid monoester **8** (295 mg, 1.1 equiv.) were placed in a Schlenk tube equipped with a stirring bar. Then, ethylene glycol (20 mL), 2-bromochalcone **7** (572 mg, 1 equiv.), and PEPPSI™-IPr (13.6 mg, 1 mol%) were sequentially added. The reaction mixture was stirred at 110 °C for 2 h, then allowed to reach room temperature and quenched with a saturated solution of ammonium chloride (NH$_4$Cl). The mixture was transferred to a separatory funnel and extracted with ethyl acetate (EtOAc) (30 mL × 3). The organic layers were dried with sodium sulfate (Na$_2$SO$_4$) and the solvent was removed in vacuo. The crude residue was purified by silica gel column chromatography with hexane/EtOAc 8:2 as eluent. The product **9** was obtained as a yellow oil (565 mg, 90%). [1]H NMR (400 MHz, CDCl$_3$) δ 7.88–7.81 (m, 1H), 7.61 (d, *J* = 7 Hz, 1H), 7.54 (d, *J* = 16 Hz, 1H), 7.57–7.51 (m, 1H), 7.49–7.42 (m, 2H), 7.42–7.35 (m, 1H), 7.33 (d, *J* = 16 Hz, 1H), 7.30 (dd, *J* = 6, 3 Hz, 1H), 7.18 (dd, *J* = 7, 1 Hz, 1H), 4.42 (m, 2H), 1.68 (t, *J* = 5, OH). [13]C NMR (101 MHz, CDCl$_3$) δ 190.7, 143.1, 141.9, 139.0, 138.7, 138.0, 133.5, 132.9, 130.9, 130.2, 130.2, 128.7, 128.6, 128.6, 128.3, 128.2, 127.8, 127.1, 123.7, 63.2. IR: 3307, 3020, 1683, 1621, 1605, 1085, 905. HRMS (ES$^+$) required for C$_{22}$H$_{18}$O$_2$: 314.1307; found (M$^+$): 314.1305.

### 3.4. Preparation of (E)-(2′-(3-Oxo-3-phenylprop-1-en-1-yl)-[1,1′-biphenyl]-2-yl)methyl Methanesulfonate (**10**)

A solution of compound **9** (314 mg, 1 equiv.) in dry methyl tert-butyl ether (MTBE) (10 mL) was cooled to 0 °C with an ice bath. Methanesulfonyl chloride (85 μL, 1.1 equiv.) and triethylamine (167.3 μL, 1.2 equiv.) were added dropwise, and the reaction mixture was stirred at 0 °C for 4 h. The reaction mixture was treated with brine before the aqueous layer was extracted with MTBE (30 mL × 3). The combined organic layers were dried with Na$_2$SO$_4$, and the solvent was removed in vacuo to provide product **10**, obtained as a pale-yellow oil (361 mg, 92%) used directly in the next step. [1]H NMR (400 MHz, CDCl$_3$)

δ 7.87 (d, *J* = 7 Hz, 3H), 7.61–7.56 (m, 1H), 7.54 (d, *J* = 7 Hz, 1H), 7.49 (d, *J* = 16 Hz, 1H), 7.51–7.46 (m, 3H), 7.45–7.40 (m, 2H), 7.36 (d, *J* = 16 Hz, 1H), 7.31–7.24 (m, 3H), 5.01 (d, *J* = 11 Hz, 1H), 4.97 (d, *J* = 11 Hz, 1H), 2.72 (s, 3H). $^{13}$C NMR (101 MHz, CDCl$_3$) δ 190.3, 142.5, 140.8, 140.6, 138.0, 133.8, 133.0, 131.5, 130.9, 130.9, 130.4, 130.2, 129.7, 128.8, 128.7, 128.7, 1282, 127.2, 123.9, 69.4, 37.7. IR: 3019, 2857, 2204, 1792, 1721, 1671, 1355, 792, 603. HRMS (ES$^+$) required for C$_{23}$H$_{20}$O$_4$S: 392.1082; found (M$^+$): 392.1083.

### 3.5. Preparation of (E)-1-((2′-(3-Oxo-3-phenylprop-1-en-1-yl)-[1,1′-biphenyl]-2-yl)methyl)quinuclidin-1-ium Mesylate (**12**)

To a solution of sulfonate **10** (392 mg, 1 equiv.) in dry tetrahydrofuran (THF) (10 mL), quinuclidine **11** (122 mg, 1.2 equiv.) was added under N$_2$ atmosphere. The reaction mixture was stirred at room temperature for 24 h. The solvent was removed under vacuum and the crude was purified by silica gel column chromatography (dichloromethane (DCM)/methanol (MeOH) 9:1). The product was obtained as a pale-yellow solid (468 mg, 93%). $^1$H NMR (400 MHz, CDCl$_3$) δ 8.02 (dd, *J* = 6, 3 Hz, 1H), 7.96 (dd, *J* = 10, 6 Hz, 3H), 7.63 (d, *J* = 9 Hz, 1H), 7.60 (dd, *J* = 9, 6 Hz, 1H), 7.53 (ddd, *J* = 16, 7, 3 Hz, 6H), 7.51 (d, *J* = 16 Hz, 1H), 7.30 (dd, *J* = 6, 3 Hz, 1H), 7.22 (dd, *J* = 6, 3 Hz, 1H), 5.05 (d, *J* = 13 Hz, 1H), 4.50 (d, *J* = 13 Hz, 1H), 3.45–3.30 (m, 6H), 2.83 (s, 3H), 2.07–2.02 (m, 1H), 1.84 (s, 6H). $^{13}$C NMR (101 MHz, CDCl$_3$) δ 189.6, 142.2, 141.2, 140.7, 137.5, 135.9, 133.4, 133.2, 132.4, 131.8, 130.8, 129.2, 129.1, 128.9, 128.5, 127.5, 125.4, 124.2, 64.5, 54.6, 39.7, 29.7, 24.0. IR: 3008, 2846, 1717, 1692, 1675, 1627, 1351, 785. m.p. 114 °C. HRMS (ES$^+$) required for C$_{28}$H$_{29}$NO$^+$ 395.2244; found (M$^+$): 395.2245.

### 3.6. Preparation of Compounds **1a** and **1b**

In a round-bottom flask equipped with a magnetic stirring bar were added, in sequence, the ammonium salt **12** (503 mg, 1 equiv.) and a 3:1 mixture of MeOH/toluene (10 mL), and the temperature was maintained at −2 °C by a cryostat bath. To this solution were sequentially added Soós' bifunctional organocatalyst *N*-[3,5-bis(trifluoromethyl) phenyl]-*N*′-[(8a,9S)-10,11-dihydro-6′-methoxy-9-cinchonanyl]thiourea (119 mg, 0.2 equiv.) and a freshly made 0.5 M aqueous sodium bisulfite solution (600 μL, 1.2 equiv.). The reaction was stirred at −2 °C overnight. Then, the organic solvent was removed *in vacuo* while the water was eliminated by freeze drying. The crude was purified by silica gel column chromatography with DCM/MeOH/Et$_3$N 98:2:5 as eluent.

3.6.1. (*S*)-1-(6′-(((1*r*,4*S*)-1l4-Azabicyclo[2.2.2]octan-3-ylium-1-yl)methyl)-2′,3′-dihydro-[1,1′-biphenyl]-2-yl)-3-oxo-3-phenylpropane-1-sulfonate **1a**

Sulfonate **1a** was obtained as white powder (171 mg, 35%). $^1$H NMR (400 MHz, CDCl$_3$) δ 8.26 (d, *J* = 8 Hz, 1H), 7.79 (d, *J* = 8 Hz, 2H), 7.63 (d, *J* = 6 Hz, 1H), 7.45 (t, *J* = 6 Hz, 2H), 7.29 (dt, *J* = 23, 11 Hz, 6H), 7.13 (d, *J* = 8 Hz, 1H), 7.01 (d, *J* = 8 Hz, 1H), 5.31 (d, *J* = 14 Hz, 1H), 4.48 (d, *J* = 14 Hz, 1H), 4.26 (d, *J* = 8 Hz, 1H), 3.99 (d, *J* = 14 Hz, 2H), 3.58 (dd, *J* = 15, 10 Hz, 1H), 3.39 (s, 3H), 3.26 (s, 3H), 2.04 (s, 1H), 1.82 (s, 6H). $^{13}$C NMR (101 MHz, CDCl$_3$) δ 197.0, 143.6, 140.3, 137.0, 136.6, 134.2, 133.7, 133.0, 130.9, 130.2, 129.2, 128.7, 128.5, 128.4, 128.3, 127.1, 125.7, 65.7, 54.5, 44.2, 24.1, 19.7. IR: 2926, 2861, 1721, 1694, 1354, 1317, 821. m.p. 225 °C. HRMS (ES$^+$) required for C$_{29}$H$_{32}$NO$_4$S: 490.2052; found (M$^+$): 490.2054.

3.6.2. (*S*)-1-(2′-(((1*r*,4*S*)-1l4-Azabicyclo[2.2.2]octan-3-ylium-1-yl)methyl)-[1,1′-biphenyl]-2-yl)-3-oxo-3-phenylpropane-1-sulfonate **1b**

Sulfonate **1b** was obtained as a white powder (171 mg, 35%). $^1$H NMR (400 MHz, CDCl$_3$) δ 7.89 (d, *J* = 7 Hz, 2H), 7.72 (d, *J* = 7 Hz, 1H), 7.63 (d, *J* = 7 Hz, 1H), 7.47 (dd, *J* = 15, 8 Hz, 1H), 7.41 (d, *J* = 7 Hz, 1H), 7.37–7.19 (m, 6H), 7.06 (d, *J* = 7 Hz, 1H), 4.79 (d, *J* = 14 Hz, 1H), 4.32 (d, *J* = 13 Hz, 1H), 4.32 (m, 1H), 4.14 (d, *J* = 15 Hz, 1H), 3.97 (dd, *J* = 18, 9 Hz, 1H), 3.34–3.19 (m, 3H), 3.15–3.00 (m, 3H), 1.97 (s, 1H), 1.75 (s, 6H). $^{13}$C NMR (101 MHz, CDCl$_3$) δ 198.6, 143.7, 140.8, 136.65, 135.8, 134.9, 134.0, 133.3, 131.7, 130.0, 128.6,

128.2, 127.9, 126.9, 124.4, 65.1, 58.8, 54.3, 40.8, 23.9, 19.6. IR: 2917, 2845, 1719, 1688, 1325, 859, 717. m.p. = 265 °C. HRMS (ES$^+$) required for $C_{29}H_{32}NO_4S$: 490.2052; found (M$^+$): 490.2053.

### 3.7. Preparation of (1s,4s)-Quinuclidin-3-yl 3-Oxobutanoate 15

Compound **15** was obtained as a pale-yellow oil following a literature procedure (200 mg, 85%) [27]. The $^1$H NMR (400 MHz, CDCl$_3$) was δ 4.87 (dd, *J* = 7, 4 Hz, 1H), 3.49 (s, 2H), 3.27 (dd, *J* = 14, 8 Hz, 1H), 2.96–2.69 (m, 5H), 2.29 (s, 3H), 2.06 (d, *J* = 3 Hz, 1H), 1.83 (tdd, *J* = 12, 6, 3 Hz, 1H), 1.71 (ddd, *J* = 19, 9, 5 Hz, 1H), 1.63–1.52 (m, 1H), and 1.42 (dt, *J* = 12, 8 Hz, 1H). The $^{13}$C NMR (101 MHz, CDCl$_3$) was δ 200.5, 166.9, 72.4, 55.2, 50.2, 47.3, 46.4, 30.3, 25.0, 24.3, and 19.3.

### 3.8. Preparation of (1R,3S,4R)-1-Benzyl-3-((3-oxobutanoyl)oxy)quinuclidin-1-ium Bromide 16

Compound **15** (105 mg, 1 equiv.) was dissolved in a mixture THF/MeOH 10:1 (5 mL), benzyl bromide (71 μL, 0.6 mmol) was added, and the reaction was refluxed for 4 h. The was solvent then removed under vacuum and the crude residue was purified by column chromatography on silica gel (DCM/MeOH 9:1). The product was obtained as an off-white solid (186 mg, 98%). $^1$H NMR (400 MHz, CDCl$_3$) δ 7.63 (d, *J* = 7 Hz, 2H), 7.49–7.38 (m, 3H), 5.16 (s, 1H), 5.05 (m, 2H), 4.34–4.25 (m, 1H), 3.96–3.82 (m, 3H), 3.73 (dt, *J* = 18, 9 Hz, 2H), 3.64 (s, 2H), 2.46 (s, 1H), 2.27 (s, 3H), 2.16–1.85 (m, 5H). $^{13}$C NMR (101 MHz, CDCl$_3$) δ 200.9, 166.5, 133.4, 130.7, 129.3, 126.7, 68.0, 66.7, 60.0, 53.8, 53.0, 49.8, 30.7, 24.5, 21.3, 18.38. IR: 2823, 1737, 1720, 1345, 1208, 1069, 788, 697. m.p. = 105 °C. HRMS (ES$^+$) required for $C_{18}H_{24}NO_3{}^+$: 302.1745; found (M$^+$): 302.1743.

### 3.9. Generation of (1R,3S,4R)-1-Benzyl-3-((3-oxobutanoyl)oxy)quinuclidin-1-ium Bromide 2

Compound **2** was not isolated, but its presence was inferred on the basis of the change in the $^1$H-NMR spectrum for compound **15** when exposed to a proton sponge. Compound **15** (76 mg, 0.2 mmol) was dissolved in CD$_2$Cl$_2$ (1 mL) and a proton sponge (1,8-Bis(dimethylamino)naphthalene) (86 mg, 0.4 mmol, 2 equiv.) was added. The solution was stirred at room temperature and, after 10 min and for 24 h, aliquots of 0.1 mL were further diluted with CD$_2$Cl$_2$ (0.3 mL) before analyzing by $^1$H-NMR spectroscopy (see Supplementary Materials).

## 4. Conclusions

In conclusion, we have reported for the first time physical evidence in support of the π-facial model [17,18] versus the superseded ionic tetrahedral model [13]. This finding stems from a revision of the X-ray data provided by Corey [12] and was carried through to the design, preparation, and analysis of model compounds **1** and **2**. The X-ray (solid state) data for compound **1** and $^1$H-NMR spectra (in solution) for compounds **1** and **2** provide definitive proof for the existence of directional $^+$N-C-H proton to RO$^-$ oxygen ion–dipole unusual H-bonding. Considering that ionic bonds are non-directional but H-bonds are directional, this study adds an important piece of information relevant to the rational design of new catalytic species and provides guidance to decipher the mechanistic details underlying the origin of enantioselectivity when enantiopure quaternary ammonium salts are employed. Hence, the data reported herein provide fundamental support to: (1) the explanation of the origin of enantioselectivity in catalyses employing quaternary ammonium salts: (2) their requirements for molecular recognition; (3) the behavior of certain materials (i.e., ionic liquids); and (4) their medicinal chemistry properties.

**Supplementary Materials:** The following supporting information can be downloaded at https://www.mdpi.com/article/10.3390/catal12070803/s1, Copies of 1H-NMR and 13C-NMR for compounds **1a**, **1b**, **2**, **7**–**12**, **15**–**16** and HMBC and HSQC of compound **16**.

**Author Contributions:** G.B. and M.W.G.-H. were responsible for carrying out the experimental work and manuscript preparation. N.S.R. was responsible for spectroscopic work; P.G., B.G.K. and M.W.G.-H. were responsible for project administration and funding acquisition. N.S.R. was

responsible for analytical data. M.F.A.A. was responsible for supervision, manuscript preparation, and conceptualization. B.T. was responsible for X-ray diffractometry. All authors have read and agreed to the published version of the manuscript.

**Funding:** This research was funded by the Irish Research Council (IRC) grant number EPSPG/2017/347; H2020-MSCA-RISE QuatSalts Grant agreement ID: 690882; and H2020-MSCA-RISE GreenX4Drug Grant agreement ID: 823939.

**Conflicts of Interest:** The authors declare no conflict of interest.

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
