# Peer review of "Quaternary Ammonium Salts Interact with Enolates and Sulfonates via Formation of Multiple +N-C-H Hydrogen Bonding Interactions"

_catalysts, doi:10.3390/catal12070803_

Round 1

Reviewer 1 Report

In this manuscript, Adamo et al. present physical evidence to confirm quaternary ammonium salts interacts with anions via a set of cooperative N+CH unusual H bonding.  

The article is well balanced with a focus on some fundamental aspects of quaternary ammonium salts. The material is novel and but not well organized and presented.

The following changes should be considered and corrected.

1. There is a dot at the end of the title. That should be removed.

2. Line 135: There is no model compound 13 and 14 in scheme 1. This should be checked. Did you mean 1a and 1b?

3. Line 151: There is no “(Scheme 3)”. Did you mean Figure 4?

4. Despite previous report of 7, its 1H NMR and 13C NMR data, like 9 and 10, should be provided in the supporting information section.

5. There is no Scheme 3 or Scheme 4 in the manuscript text, as claimed in line 151 and 185.

6. more literature reference should be given in the introduction section. Please consider following article and references therein. 10.1002/chem.200903016

It is suggested that the authors double check their figure, scheme and molecule numbering method in this manuscript.   

Author Response

Dear Reviewer, 

many thanks for your kind review of our manuscript. We have listed each of your comments below with responses in red.  A substantial review of the document has also been performed with key sections reworded to correct for English and to ensure clarity of sentiment. 

The following changes should be considered and corrected.

1. There is a dot at the end of the title. That should be removed.

This has been corrected.

2. Line 135: There is no model compound 13 and 14 in scheme 1. This should be checked. Did you mean 1a and 1b?

Your suggestion was correct and the error has been corrected.

3. Line 151: There is no “(Scheme 3)”. Did you mean Figure 4?

This was an error - it should have read "(Scheme 1)". This has been corrected. 

4. Despite previous report of 7, its 1H NMR and 13C NMR data, like 9 and 10, should be provided in the supporting information section.

The NMR spectrum of compound 7 has been included.

5. There is no Scheme 3 or Scheme 4 in the manuscript text, as claimed in line 151 and 185.

This error has been acknowledged and corrected. 

6. more literature reference should be given in the introduction section. Please consider following article and references therein.10.1002/chem.200903016

The suggested reference has been included.

It is suggested that the authors double check their figure, scheme and molecule numbering method in this manuscript.   

This has been done.

We trust that the corrections will be to the reviewers satisfaction. 

With regards,

Reviewer 2 Report

-          At Introduction, specify the sectors of the industry were these materials can be used.

-          Authors mention (line 135): “we synthesized model compounds 13 and 14 (Scheme 1)”, but these compounds are not numbered in the Scheme 1. Please, correct the Scheme 1.

-          The Chapter 3 is not mentioned in the manuscript. Please check the chapter numbering.

-          At chapter 4, mention the full name for all reagents (e.g: CDCl3, i-Pr PEPPSI, EtOAc, MTBE)

-          English improvement is required. Some examples but not all are as the following:

        - At 1.Introduction, line 52, modify “p-NO2-phenolate” with “p-NO2-phenolate”

        - At 2, lines 82, 95, modify “-NO2” with “-NO2

        - At 2, lines 187, modify CD2Cl2” with “CD2Cl2

Author Response

Dear Reviewer, 

many thanks for your kind review of our manuscript. We have listed each of your comments below with responses in red.  A substantial review of the document has also been performed with key sections reworded to correct for English and to ensure clarity of sentiment. 

-          At Introduction, specify the sectors of the industry were these materials can be used.

This has been done.

-          Authors mention (line 135): “we synthesized model compounds 13 and 14 (Scheme 1)”, but these compounds are not numbered in the Scheme 1. Please, correct the Scheme 1.

This has been corrected.

-          The Chapter 3 is not mentioned in the manuscript. Please check the chapter numbering.

This has been done.

-          At chapter 4, mention the full name for all reagents (e.g: CDCl3, i-Pr PEPPSI, EtOAc, MTBE)

This has been done.

-          English improvement is required. Some examples but not all are as the following:

        - At 1.Introduction, line 52, modify “p-NO2-phenolate” with “p-NO2-phenolate”

This has been done.

        - At 2, lines 82, 95, modify “-NO2” with “-NO2

This has been done.

        - At 2, lines 187, modify CD2Cl2” with “CD2Cl2

This has been done.

We trust that the revisions made are to the satisfaction of the reviewers. 

With regards,

Round 2

Reviewer 1 Report

no comment. 

Reviewer 2 Report

Dear Sirs,

The manuscript was improved and it can be publish in this form.